# Risk Assessment Model System for Aquatic Animal Introduction Based on Analytic Hierarchy Process (AHP)

**DOI:** 10.3390/ani13122035

**Published:** 2023-06-19

**Authors:** Xuxin Zhang, Hehe Du, Zhouzhou Zhao, Ying Wu, Zhenjie Cao, Yongcan Zhou, Yun Sun

**Affiliations:** 1Sanya Nanfan Research Institute, Hainan University, Sanya 572022, China; zhangxuxin2017@163.com (X.Z.);; 2Collaborative Innovation Center of Marine Science and Technology, Hainan University, Haikou 570228, China; duhehe1005@163.com (H.D.);; 3Hainan Provincial Key Laboratory for Tropical Hydrobiology and Biotechnology, College of Marine Science, Hainan University, Haikou 570228, China

**Keywords:** biological invasion, invasive species (IS), introduced aquatic animals, risk assessment, analytic hierarchy process (AHP), aquatic ecosystem

## Abstract

**Simple Summary:**

Invasive species (IS) can upset both aquatic and terrestrial ecosystems which, in turn, can negatively affect production and economic development. Thus, many countries understand the importance of performing risk assessments on the potential for species to invade an ecosystem. Here, we report on our development of a quantitative risk assessment model to estimate the introduction of invasive aquatic animals into China’s inland waters. This model is based on the analytic hierarchy process (AHP). We propose that the use of this model can provide the basis to better understand the ecological impact of invasive aquatic animals and also effective protocols for risk management should an invasion take place.

**Abstract:**

The spread of invasive species (IS) has the potential to upset ecosystem balances. In extreme cases, this can hinder economical utilization of both aquatic (fisheries) and terrestrial (agricultural) systems. As a result, many countries regard risk assessment of IS as an important process for solving the problem of biological invasion. Yet, some IS are purposefully introduced for what is seen as their potential economic benefits. Thus, conducting IS risk assessments and then formulating policies based on scientific information will allow protocols to be developed that can reduce problems associated with IS incursions, whether occurring purposefully or not. However, the risk assessment methods currently adopted by most countries use qualitative or semiquantitative methodologies. Currently, there is a mismatch between qualitative and quantitative assessments. Moreover, most assessment systems are for terrestrial animals. What is needed is an assessment system for aquatic animals; however, those currently available are relatively rudimentary. To fill this gap, we used the analytic hierarchy process (AHP) to build a risk assessment model system for aquatic IS. Our AHP has four primary indexes, twelve secondary indexes, and sixty tertiary indexes. We used this AHP to conduct quantitative risk assessments on five aquatic animals that are typically introduced in China, which have distinct biological characteristics, specific introduction purposes, and can represent different types of aquatic animals. The assessment results show that the risk grade for *Pterygoplichthys pardalis* is high; the risk grade for *Macrobrachium rosenbergii*, *Crassostrea gigas*, and *Trachemys scripta* elegans is medium; and the grade risk for *Ambystoma mexicanum* is low. Risk assessment of the introduction of aquatic animals using our AHP is effective, and it provides support for the introduction and healthy breeding of aquatic animals. Thus, the AHP model can provide a basis for decision-making risk management concerning the introduction of species.

## 1. Introduction

Biological invasions are considered a major driving factor in the vanishing of biodiversity [1]. Invasive species (IS) can cause serious damage to an ecosystem, which can lead to negative impacts on human society; these impacts include economic losses caused by trade and agricultural production obstacles and threats to human health [2,3]. An IS is a non-native species that has invasive biological characteristics (such as strong adaptability and spread capability) or may have negative impacts on ecology, agriculture, fisheries, food, or human health, including artificially introduced and unintentionally arrived nonindigenous species [4,5,6]. Another definition of IS is a non-native species that can have harmful effects on the economy, environment, or human health (supported by the International Union for Conservation of Nature (IUCN)) [7]. An IS affects the normal functioning of ecosystems through mechanisms such as predation, hybridization, and competition. An IS can also have indirect impacts, such as reducing the abundance of local species [8]. Although there are very few ecosystems in the world that are free of introduced species, an increasing proportion of biomes, ecosystems, and habitats are being impacted by IS [9].

Once the ecological tipping point has been passed, the impact on ecosystems by IS may be irreversible [10]. Many IS incursions are because of human activities [11], such as overexploitation of local biological resources, religious beliefs, and shipping trade. These have significantly altered biodiversity and community structures on a global scale; the result of this is to add serious socioeconomic burdens, as well as losses of ecosystem services [12,13,14,15]. Large-scale species migration and biological invasion can initiate sharp declines in the number of native species and the introduction of unknown pathogens. Some inductions of IS have been for the purpose of economic development (agricultural production, trade, tourism, etc.); this has become increasingly frequent in recent decades and, as a result, are emerging as a significant threat to global biodiversity and economic losses [9,16,17,18].

To address the problem of biological invasion caused by the introduction of IS into agricultural, some developed countries (e.g., EU region) have utilized risk assessment (RA) as an important measure to strengthen the risk analysis of non-native species invasion [11,19]. Risk analysis methodology was originally developed by the nuclear and aerospace industries to identify the likelihood of hazards. Thus, the scheme has gradually been applied to biological research [20]. Risk assessments are a fundamental part of risk analysis methodology. By conducting risk assessments on the entry, exposure, and consequences of introduced species and developing scientific introduction strategies based on the assessment results, the harm caused by IS can be effectively reduced [21]. However, upon reviewing risk assessment records from various countries around the world, we found that most of the assessment outcomes were based on the results of qualitative analysis obtained from monitoring reports of introduced species or outbreak records of related diseases in the species-exporting countries [22]. Compared with quantitative analysis, qualitative analysis cannot accurately address what actually happens to agricultural production. Thus, that assessment method has limitations, is ambiguous, is less accurate, and is less universal than quantitative analysis. In addition, quantitative analysis has the unique advantages of high testability, repeatability, comparability, and transparency [23].

The analytic hierarchy process (AHP) was proposed by the American operational research scientist Saaty in the 1970s. AHP skillfully combines quantitative analysis with qualitative analysis and has a wide range of application in the field of comprehensive evaluation [24]. After more than 40 years of research and development, the AHP has become one of the most mainstream multiple criteria decision-making (MCDM) analysis methods, and its main advantages include simplicity, flexibility, and rigorous and strong operability [25]. Currently, AHP is widely used in animal production, resource utilization, natural disaster prevention and control, and other fields: Hadi Veisi proposed the application of the AHP in the multicriteria selection of agricultural irrigation systems [26]; Priyanka Yadav established a decision support system for the selection of biogas upgrade technologies based on the AHP [27]; and Mehmet Cihan Aydin combined geographic information systems with the AHP to assess the flood risk in Bitlis Province, Turkey [28]. Moreover, according to the various characteristics and objects of analysis, the AHP encompasses many different types, such as the hesitant analytic hierarchy process, the fuzzy analytic hierarchy process, and the sparse analytic hierarchy process [29,30,31].

On a global scale, aquatic ecosystems are an important component in many regions; they provide many ecosystem services and, of course, are critical to modern aquaculture and animal husbandry [32,33]. Human-mediated introductions of aquatic animals with the purpose of improving agricultural economic development (either recreational and/or aquaculture) can alter the species composition of aquatic ecosystems and thereby change ecosystem functioning (the Japanese archipelago is an example) [34]. At the same time, these changes may have widespread consequences. For example, the opening of the Suez Canal has significantly changed the diversity of fish in the Mediterranean region, which has had an impact on Italy’s socioeconomic and human health [35]. With the continual expansion of these changes, aquatic bio-invasions have caused environmental issues worldwide, influencing the structure and function of water ecosystems globally, as well as imposing serious socioeconomic burdens [36,37]. Because of the more complex composition of aquatic ecosystems and their distinct differences from terrestrial ecosystems, there are significant differences in the management of non-native species between aquatic ecosystems and terrestrial ecosystems [38,39]. The negative impacts of IS on aquatic ecosystems (community composition, organic matter concentration, and water turbidity) tend to be worse after the invasion [40]. Currently, many risk management efforts regarding IS are aimed at controlling the invasion of terrestrial animals, while, by comparison, related research on risk management of aquatic ecosystems (such as marine ecosystems) is lagging behind [41]. Here, we established a quantifiable risk assessment model system for the introduction of aquatic animals on the basis of the analytic hierarchy process (AHP) using China as the proposed importing country. We conducted quantitative risk assessments for five commonly introduced aquatic animals (including fish, crustaceans, shellfish, and amphibians) in southern China and ranked their risk levels based on the assessment results. Thus, the main objectives of the study were: (1) propose a risk assessment model system for the introduction of aquatic animals that is compatible with both species’ own risk, as well as the transmission disease risk for reference and use by other countries in the world; (2) optimize the existing qualitative or semiquantitative risk assessment system for aquatic animal introduction on the basis of the actual introduction of aquatic animals and relevant policies and regulations; (3) fill the gaps in the research related to the management of non-native species in aquatic ecosystems.

## 2. Materials and Methods

### 2.1. Index Composition of Risk Assessment Model System

If we want to carry out a specific quantitative risk assessment of an introduced non-native species, we must consider both the biological risk of the species itself and the risk of epidemic diseases (including epidemic diseases of wild and domestic aquatic animals and zoonotic diseases of aquatic animals) [42]. The biological risk of the species itself needs to be assessed on whether the introduction of the species will threaten indigenous species and their habitats, as well as the existing ecosystem and environment of the introduction site. Additionally, it is also important to consider whether relevant policies and regulations can provide a sufficient guarantee for safe species introduction [43,44,45]. The risk of infectious diseases should include a risk assessment of pathogen exposure; hazards to indigenous species (including becoming direct victims and pathogen vectors); consequences of disease outbreak; impacts on human health, social economy, and ecological environment; and other relevant factors [46,47,48]. Regarding the internationally advanced risk assessment system, we determined the indices of the risk assessment model system for the introduction of aquatic animals (see Table 1).

### 2.2. Calculation of Index Weights at All Levels

One of the core steps of the AHP is to form accurate pairwise comparison matrices defined by the users [49]. In this study, the steps to calculate the weight values were as follows: (1) first, we determined the decision-making objective, structured the decision hierarchy, and constructed the objectives from a broad perspective from the intermediate level to the alternative level; (2) then, we scored the importance of each peer-level index, discussed and summarized the scoring results, constructed pairwise comparison matrices, calculated them separately, and obtained the weight values; (3) finally, after calculating the weights, we verified the rationality of the weight coefficient distribution using MATLAB software to calculate the maximum eigenvalue of the judgment matrix, λ max, and the consistency index, *CI*. We then judged whether the distribution of the weight coefficient was reasonable (<0.1 is reasonable) according to the result obtained via the quotient of the average random consistency index, *RI*, and the consistency index, *CI* (the random consistency ratio) using the mathematical formula as follows: *CI* = (λ max − n)/(n − 1); *CR* = *CI*/*RI* (n is the number of indices selected for each criterion layer, the calculation results were carried to four decimal places) [50,51,52].

The relationships and roles of risk assessment indices are different. According to their contribution, they can be divided into multiple relationships, cumulative relationships, and substitution relationships. When the indices in the system are independent of each other and contribute independently to the value of the indices at the next level, their relationship is cumulative. The relationships among indices at all levels of this model system are cumulative. According to the calculation, the final function expression is: 

Risk of introduction of imported aquatic animals (R): R = 0.3873R1 + 0.1397R2 + 0.2748R3 + 0.1982R4.

Hazard assessment of introduced species (R1): R1 = 0.6667P1 + 0.3333P2;
P1 = 0.6667p11 + 0.3333p12;
P2 = 0.0866p21 + 0.2150p22 + 0.0433p23 + 0.3408p24 + 0.2544p25 + 0.0599p26.

Entry assessment (R2): R2 = 0.0883P3 + 0.4824P4 + 0.2718P5 + 0.1575P6;
P3 = 0.0449p31 + 0.1760p32 + 0.1760p33 + 0.1803p34 + 0.0395p35 + 0.0960p36 + 0.0960p37 + 0.0606p38 + 0.1307p39;
P4 = 0.2488p41 + 0.5502p42 + 0.0826p43 + 0.1184p44;
P5 = 0.1402p51 + 0.2504p52 + 0.2055p53 + 0.1346p54 + 0.0710p55 + 0.0670p56 + 0.0633p57 + 0.0322p58 + 0.0358p59;
P6 = 0.1572p61 + 0.4596p62 + 0.2945p63 + 0.0887p64.

Exposure assessment (R3): R3 = 0.3759P7 + 0.1321P8 + 0.0867P9 + 0.4053P10;
P7 = 0.3250p71 + 0.1251p72 + 0.1937p73 + 0.3562p74;
P8 = 0.3338p81 + 0.5907p82 + 0.0755p83;
P9 = 0.5246p91 + 0.0918p92 + 0.3337p93 + 0.0499p94;
P10 = 0.1646p101 + 0.2792p102 + 0.3916p103 + 0.1646p104.

Consequence assessment (R4): R4 = 0.8000P11 + 0.2000P12;
P11 = 0.0602p111 + 0.1001p112 + 0.1767p113 + 0.3410p114 + 0.3220p115;
P12 = 0.3836p121 + 0.1918p122 + 0.1918p123 + 0.0708p124 + 0.0558p125 + 0.1062p126.

### 2.3. Quantitative Evaluation Criteria and Basis of Indices

The most prominent feature of quantitative assessment is that it provides a numerically defined threshold value of an empirically measurable quantity for each assessment index [23], which solves the problem of incommensurability among indices; clear evaluation criteria and bases can also eliminate (or reduce) the bias in evaluation results caused by subjective factors and increase the correctness and accuracy of the evaluation results. Our system divides the risk value of each tertiary index into six levels; that is, the risk is divided from low to high as follows: 0 (negligible risk), 1 (low risk), 2 (slight risk), 3 (medium risk), 4 (high risk), and 5 (extremely high risk), each tertiary index has the same assignment range. Users can score each index according to the literature and expert opinions. For the indices that need to be discussed and scored in the tertiary indices, in order to eliminate the deviation of assessment results caused by subjective factors, we designed more specific assignment situations and scoring evaluation rules. Users can discuss the indices, search for matching assignment situations in the evaluation rules, and score them on the basis of the actual situation. Refer to Table 2, Table 3, Table 4 and Table 5 for the specific assessment criteria and evaluation bases and Table 6 for the specific situation and scoring evaluation rules of the indices to be discussed.

### 2.4. Assessment of Introduction Risk (R): Setting of Risk Grade

We referred to the current international pest grading system and the management of non-native species grading, consulted the relevant literature and the grading method in the text, and combined the specific situation of the risk assessment model system to set the risk level as follows: when 0 < R ≤ 1, the risk is negligible, and the introduction can be carried out; when 1 < R ≤ 2, the risk is low, and the introduction can be carried out if the national policy allows; when 2 < R ≤ 3, the risk is medium, and limited introduction can be carried out, but strict management measures must be taken; when 3 < R ≤ 4, the risk is high, and introduction is not recommended; if the introduction is required, close communication customs and other relevant departments must be conducted, and strict introduction strategies and risk prevention and control measures must be developed; when 4 < R ≤ 5, the risk is extremely high, and introduction cannot take place.

### 2.5. Verification of the System’s Correctness and Rationality with Risk Assessment Examples

To verify the correctness and rationality of the above risk assessment system used in the actual introduction of aquatic animals, we selected five introduced aquatic animals cultivated in southern China, *Pterygoplichthys pardalis*, *Macrobrachium rosenbergii*, *Crassostrea gigas*, *Trachemys scripta elegans*, and *Ambystoma mexicanum*, for use in our model for the introduction risk assessment. If the assessment results are consistent with international assessment systems currently being used, and if they comply with the reference materials in the germplasm resource databases of most countries, it can be confirmed that the assessment results are realistic and effective, and the assessment system is rational.

## 3. Results and Analysis

### 3.1. Risk Assessment Examples: Evaluation of Indices at Different Grades

In this assessment, we used the following methods to score various indices: literature reviews; consultation with experts; participation in the work of relevant units such as the Ministry of Ecology and Environment, Ministry of Natural Resources, and China Customs; observation of farms and aquariums with target species; and communication with aquaculture-related personnel. Table 7 shows the scoring results of each tertiary index of risk assessment. (Among these tertiary indices, some indices required discussion before scoring. See Appendix A for details on these indices.)

### 3.2. Calculation of Indices and Analysis of Assessment Results

According to the scoring of the tertiary index discussed in Section 3.2, we calculated the values of the secondary indices (P1–P12), primary indices (R1–R4), and total indices (R) of each introduced aquatic animal (see Table 8 for details).

According to the calculation, the order of the risk grade from large to small is as follows: Pterygoplichthys pardalis > Macrobrachium rosenbergii > Crassostrea gigas > Trachemys scripta elegans > Ambystoma mexicanum. The risk grade of Pterygoplichthys pardalis is high; if introduction is necessary, a scientific introduction strategy and risk prevention and control measures must be employed, and limited introduction must be carried out with the close cooperation of the customs and quarantine departments and other relevant units; the risk grades of Macrobrachium rosenbergii, Crassostrea gigas, and Trachemys scripta elegans are medium, so they can be introduced in a standard manner under strict management measures; the risk grade of Ambystoma mexicanum is low, and it can be introduced under the conditions permitted by the importing country’s national policy. 

The assessment results are highly consistent with the assessment results of widely used international assessment systems and reference materials in germplasm resource databases of China and other countries.

## 4. Discussion

### 4.1. Discussion and Analysis of Assessment Results

The following results were obtained in this assessment. (1) The risk grade of *Pterygoplichthys pardalis* is high. *Pterygoplichthys pardalis*, commonly known as the armored catfish and first used as an aquarium pet, is an IS that endangers many aquatic ecosystems around the world. However, because of its wide-ranging tolerance, strong reproductive capacity, omnivorous feeding habits, and other characteristics, *Pterygoplichthys pardalis* has now established large wild populations in most rivers around the world, seriously affecting agricultural production and ecological environments of introduction areas. Moreover, *Pterygoplichthys pardalis* is a direct or indirect host of various pathogens, and natural spread caused by improper management can lead to unknown disease outbreaks, endangering local populations [53,54,55]. (2) The risk grade of *Macrobrachium rosenbergii* is medium; *Macrobrachium rosenbergii* represent a large category of aquaculture, namely, crustaceans, which have the potential not only to suppress the growth of local shrimp populations but also to become hosts (or intermediate hosts) to a variety of extremely dangerous pathogens [56]. Therefore, strict inspection and quarantine must be conducted during introduction, while ensuring that intermediate links, such as transportation and monitoring, can achieve the desired results. During breeding, it is necessary to optimize the layout of the breeding structure, improve awareness of disease prevention and control, and strengthen breeding management [57], such as controlling the physicochemical and biological factors in the cultivation process to improve immune resistance and reduce the risks to a controllable range [58]. (3) The risk grade of *Crassostrea gigas* is medium; as a shellfish widely introduced and cultivated around the world, *Crassostrea gigas* has high nutritional value and market demand [59,60]. *Crassostrea gigas* have strong adaptability and rapid growth performance, and shellfish are easier to manage in quarantine and transportation than fish and crustaceans. During the introduction and cultivation process, strengthening the inspection and control of diseases—as well as strengthening the management of *Crassostrea gigas* in various fisheries to avoid escape and dispersion during cultivation—can reduce the risk of *Crassostrea gigas* to a certain extent [61,62]. (4) The risk grade of *Trachemys scripta elegans* is medium. Although *Trachemys scripta elegans* is considered one of the 100 worst IS according to the IUCN and drives native freshwater turtles to consume suboptimal resources [63,64], the degree of control of non-native species is generally higher than that of pathogens. It is necessary to strengthen the supervision of the introduction and cultivation of *Trachemys scripta elegans*; prevent escape, diffusion, and random breeding; and strictly manage the areas where the *Trachemys scripta elegans* population settles, as *Trachemys scripta elegans* can coexist harmoniously with native species [65]. (5) The risk grade of *Ambystoma mexicanum* is low. *Ambystoma mexicanum* is a species protected by the Convention on International Trade in Endangered Species of Wild Fauna and Flora (CITES) [66]; although artificial cultivation has been achieved in some countries, for most countries or regions around the world, the wild *Ambystoma mexicanum* resource is still scarce. Artificially cultivated *Ambystoma mexicanum* can be reasonably sold according to national policies and regulations, while for wild resources, it is necessary to select suitable microhabitats (such as places with suitable temperatures and rich vegetation) and establish natural reserves to strengthen the protection of wild resources and prevent possible risks that affect *Ambystoma mexicanum* [67,68].

### 4.2. The Value and Significance of the Risk Assessment Model System in the Introduction of Non-Native Species

The introduction of non-native species is usually aimed at promoting economic development, but many other factors should also be considered, such as food safety, policies and regulations, and introduction costs [69,70,71], as these factors are as important as economic development. An excellent and practical risk assessment model system should cover the discussion and exploration of the above factors. At the same time, as the basis for many international accounting standards policies and decisions related to biological invasion, the repeatability and reliability of risk assessments are crucial. Both repeatability and reliability of risk assessment systems are prerequisites for obtaining accurate and valuable results. The value of assessment results is a key factor in measuring whether the modeling system is meaningful. Assessment results can indicate the direction for the introduction of non-native species, provide practical suggestions for management of non-native species, and directly reflect the value of the evaluation results and, even more importantly, the significance of the existence of the risk assessment model system.

Our risk assessment model system is proposed on the basis of the previous risk assessment schemes and has several key advantages: (1) The model integrates two major directions: ecological risk of the species itself and risk of transmission of unknown diseases. It covers the biological characteristics of the species (and pathogens) while taking into account the economic benefits, policies and regulations, ecological environment, and other factors that directly affect the introduction of non-native species. (2) The model solves the problem of incommensurability among indices through quantitative scoring, transforming complex issues that cannot be described clearly into intuitive, digitally presented assessment results. (3) The lack of authentic and reliable evidence for non-native species is a key factor that reduces the accuracy of the assessment results. In our assessment system, there will be expert discussions of indices related to this part, and users can grade the indices on the basis of the experts’ discussions and suggestions, supported by solid theoretical bases, to ensure the accuracy of the scoring results. In addition, expert discussions can also serve as a basis for introduction strategies and management measures of non-native species. (4) Compared with other risk assessment systems of non-native species in the world, this system is more suitable for economic aquatic animals, and the indices involved in the model system can better reflect the problems that may be exposed during the actual introduction process. Therefore, when formulating introduction strategies and management measures, it will be more intuitive.

One limitation of our risk assessment model system is that the main assessment objects of this assessment system are non-native aquatic animals. Its core purpose is assessing the benefits regarding economic development (such as those for cultured fish and ornamental fish for aquatic plants), as well as assessing other aquatic animals introduced for various purposes (such as medical treatment, ecological environment improvement, and restoration), and the compatibility between this assessment system and these last three objectives is not high. Another constraint is that the assessment system focuses on pathogens that may be carried by introduced species for the risk assessment of infectious diseases, making the system unlikely to be suitable for assessing other potentially introduced pests (such as harmful plants and insects that may sneak in with the introduced species). These issues will be addressed in the further development of the model.

### 4.3. Application of Risk Assessment in Non-Native Species Management

Over the past several decades, there has been enormous growth in research interest regarding IS [72]. With the gradual deepening of research, scientists have also proposed some widely used model systems for the risk assessment of non-native species. For example, the Fish Invasiveness Screening Kit (FISK), widely used in Europe, is adapted for freshwater fishes based on the Australian Weed Risk Assessment (WRA). The FISK scoring system contains 49 question items that include species biogeography/history, as well as biological characteristics and ecological attributes that represent invasiveness. On the basis of the assessment and scoring results, non-native fish can be classified according to the potential risk [73]. At present, FISK is still continually updated and widely used [74].

Cynthia S. Kolar and David M. Lodge proposed a science-based simple risk assessment protocol named the “Fish Invasion Screening Test” (FIST); the invasiveness screening criteria of FIST include screening for potential biological features, such as growth, culture level, history of establishment, breeding in the wild, phenotypic plasticity, ability to live off a wide range of food types, competition with local species, diseases, dispersal ability (propagule pressure), and other characteristics attributable to invasiveness [69,75]. With the support of databases (such as DIAS and GISD), this risk assessment system can yield objective and correct assessment results.

On the basis of a risk analysis program for non-native species in aquaculture in Europe used to screen all animals and plants in aquatic ecosystems, the Aquatic Species Invasiveness Screening Kit (AS-ISK) combines the recent EU regulations on the prevention and management of the introduction and spread of IS. This not only includes species types of fresh, brackish, and marine waters but also provides increasing comparability across aquatic species [76]. Using AS-ISK to evaluate non-native aquatic species (including both existing and potential future non-native species), the assessment system can make the process more flexible and improve the accuracy of the assessment results, as it incorporates analyses of policies and regulations and enables comparison among different aquatic species.

The Integrated Biosafety Risk Assessment Model (IBRAM) is a model framework that can be used for evaluating the risks of imported products harboring IS. The IBRAM framework consists of multiple interrelated models that describe the entry of pests into the country, their escape along trade pathways, their initial dispersal into the environment, their habitat suitability, the probabilities of their establishment and spread, and the consequences of these intrusions. Compared with the previously discussed assessment system, the assessment object of IBRAM usually concerns imported products harboring IS, and the assessment focuses on the risk of these organisms establishing and dispersing within the region of assessment, which can better integrate with the actual situation of trade [77].

### 4.4. Relationship between Risk Assessment and Non-Native Species Management

The main reason for the migration or invasion of non-native species is that the intentional or unintentional actions of humans cause their own individuals or reproductive bodies to spread beyond the limits of the normal geographic regions to which they originally belonged [72]. We need to recognize that the impacts of the introduction of an non-native species on factors such as ecosystem function, species composition, and species richness cannot be dichotomized simply as beneficial or harmful; even if the introduced species are IS, their relationships with indigenous or abiotic environments will comprise positive and negative interactions [78]. Moreover, there are significant differences in the types, uses, and transmission routes of various non-native species, which leads to their complex relationships with ecological environments, social economies, policies, regulations, customs, and other factors [79]. Risk assessment should serve as a tool for the management of non-native species that integrates the above factors and can objectively evaluate the positive and negative effects of non-native species, providing decision-making recommendations and bases for the management of IS—providing scientific management solutions for IS that already exist in the region, predicting the invasion mechanisms and conditions of non-native species that may become IS, providing valuable introduction strategies for incoming non-native species [80], and concurrently judging and predicting the feasibility and effectiveness of various management schemes and control programs for established invasive populations [81]. Therefore, the relationship between risk assessment and non-native species management (with the emphasis on risk management) is interactive [82].

With the continual development of agricultural science and technology, increasingly advanced technologies are being applied to risk assessment and the management of non-native species, such as using unmanned aerial vehicles and satellites to detect non-native species [83] and establishing new marine protected areas [84]. At the same time, with the gradual and in-depth development of aquatic ecosystems, risk assessment, and non-native species management are also effectively compatible with other resource utilization projects, such as the development of electric fields, wind energy, and tidal energy.

## 5. Conclusions

The global trade of agricultural products has gradually entered an age of popularization and diversification, and biological invasion will be a major obstacle to the development of aquaculture, animal husbandry, and forestry. The prevention, control, and management of IAS is a difficult task for any country in the world. As the foundation of risk management for non-native species, risk assessment is also an important basis for policies and regulations for preventing and controlling IAS in many countries around the world; it plays an irreplaceable role in the field of controlling biological invasion. The risk assessment model system for the introduction of aquatic animals proposed in this study conducts, via quantitative analysis, a risk assessment on the introduction of aquatic animals, to formulate more scientific introduction strategies and more effective risk management measures by using the scoring results and overall assessment results of various indices. For IS that already exist in an area, specific control measures (capture, prevention, etc.) should be implemented for those species under the conditions permitted by policies and regulations. For non-native species that may become IS in an area, it is necessary to immediately communicate with the fishery department or other relevant units and carry out strict monitoring to control the population size. For new varieties that have never been introduced in the area, risk assessments should be conducted first to determine the potential risks and propose possible impacts, employing strict introduction strategies and risk management measures before introduction.

## Figures and Tables

**Table 1 animals-13-02035-t001:** The indices of the risk assessment system for the introduction of aquatic animals.

Primary Index	Secondary Index	Tertiary Index
Hazard assessment of introduced species (R1)	Basic attributes of introduced species (P1)	Basic invasive situation of non-native species (p11)
		Basic endangered situation of non-native species (p12)
	Self-hazard of introduced species (P2)	Suitability of environmental factors (temperature, dissolved oxygen, salinity, pH, etc.) (p21)
		Natural enemies of introduced species (p22)
		Feeding habits of introduced species (p23)
		Impact of introduced species on indigenous aquatic animals in receiving waters (p24)
		Impact of introduced species on abiotic environment of receiving waters (p25)
		Impact of introduced species on biodiversity (mainly referring to the impact on algae, aquatic plants, microorganisms, etc.) (p26)
Entry assessment (R2)	Official fishery and medical management systems of both countries (P3)	Trade relations between exporting and importing countries (p31)
		National fisheries and medical administrations and their responsibilities (p32)
		Local fisheries and medical management organizations and their responsibilities (p33)
		Aquatic animal health laws and regulations system in importing country (p34)
		Stakeholder obligations (p35)
		Status of nationally recognized fishery medical diagnostic laboratories (p36)
		Status of nationally recognized aquatic animal epidemic laboratories at all levels (district level, city level, and provincial level) (p37)
		Accreditation and management of national reference animal laboratories (p38)
		National animal health outlay support in importing country (p39)
	Diseases carried by introduced species (P4)	Type and quantity of diseases carried by introduced species (p41)
		Risk degree of diseases carried by introduced species (p42)
		Disease epidemic situation in exporting countries in the past five years (p43)
		The importance attached to the disease by exporting countries (p44)
	Introduction species epidemic prevention and control system (P5)	Notification of aquatic animal epidemics (p51)
		Occurrence and disposal situation of epidemics (p52)
		Management measures for specific epidemic regionalization (biosafety isolation area) (p53)
		Animal epidemic monitoring plan and implementation status (p54)
		Vector monitoring plan and implementation status (p55)
		Immunization of aquatic animal diseases (p56)
		Identification and traceability of aquatic animals (p57)
		Aquaculture enterprise registration and biosafety requirements (p58)
		Quarantine measures at coastal ports (p59)
	Economic efficiency of introduced species (P6)	Number and scale of introduced species (p61)
		Economic benefits from introduction of species (p62)
		Costs of introducing species for quarantine (p63)
		Costs of intermediate links (monitoring, detection, transportation, etc.) (p64)
Exposure assessment (R3)	Biological characteristics of pathogenic organisms (P7)	Basic characteristics of the pathogens (p71)
		Type and quantity of hosts (p72)
		Situation of pathogen infection, morbidity, and mortality (p73)
		Threat of pathogens to humans (p74)
	Spread of pathogenic organisms (P8)	Transmission vector and route of pathogens (p81)
		Transmission speed and capacity of pathogens (p82)
		Impact of abiotic factors on pathogen transmission (p83)
	Host infection symptoms and pathological changes (P9)	Host infection symptoms (p91)
		Pathogenic infection site (p92)
		Pathological changes of pathogenic infection (p93)
		Specific situation of the victim host (age, sex, etc.) (p94)
	Detection methods and prevention and control measures (P10)	Status of vaccines (p101)
		Pathogen detection (diagnosis) methods (p102)
		Pathogen prevention measures (p103)
		Pathogen treatment methods (p104)
Consequence assessment (R4)	Direct consequences (P11)	Epidemic infection rate of aquatic animals (p111)
		Epidemic incidence rate of aquatic animals (p112)
		Epidemic mortality rate of aquatic animals (p113)
		Impacts on human health (p114)
		Direct impacts on economy (p115)
	Indirect consequences (P12)	Costs and difficulty of monitoring, prevention, and control (p121)
		Potential transaction losses (p122)
		Potential impacts on social economy (p123)
		Adverse impacts on ecological environment (p124)
		Hazards to biodiversity (p125)
		Costs of ecological environment restoration (p126)

**Table 2 animals-13-02035-t002:** Assessment criteria for the hazard assessment of introduced species.

Primary Index	Secondary Index	Tertiary Index	Assessment Criteria and Evaluation Bases
			0 (Negligible Risk), 1 (Low Risk), 2 (Slight Risk), 3 (Medium Risk), 4 (High Risk), 5 (Extremely High Risk)
R1	P1	p11	Based on the list of IS in the importing country and the actual situation of non-native species, make a basic judgment on the invasion situation of the introduced species and assign a grade according to the invasion risk of introduced species: 0 (negligible risk), 1 (low risk), 2 (slight risk), 3 (medium risk), 4 (high risk), or 5 (extremely high risk).
		p12	Based on the IUCN Red List of Threatened Species and the actual situation of the importing country, make a basic judgment on the endangered situation of the introduced species and assign a grade according to the risk of introduced species: 0 (negligible risk), 1 (low risk), 2 (slight risk), 3 (medium risk), 4 (high risk), or 5 (extremely high risk).
	P2	p21	(0) The introduced species can hardly survive in the external environment. (1) It is difficult for the introduced species to survive in the new environment. (2) The external environment basically conforms to the survival conditions of the introduced species. (3) The external environment basically conforms to the survival conditions as well as the reproduction conditions. (4) Both survival and reproduction conditions are satisfied, and the external environment is relatively suitable. (5) The external environment is very suitable for the introduction of species to survive and reproduce.
		p22	(0) There are many powerful natural enemies in the region, and the introduced species can survive only under specific protection. (1) Various natural enemies pose a great threat to the survival of the introduced species. (2) There is a certain predatory or competitive relationship between the introduced species and natural enemies in the region, and the natural enemies prevail. (3) The introduced species have an obvious predatory or competitive relationship with the natural enemies in the region, and the relationship between them is close. (4) The number of natural enemy species in the region is relatively small, which leads to the long-term dominance of the introduced species. (5) There are no effective natural enemies in the region.
		p23	Discuss the feeding habits of the introduced species and assign a grade according to the potential risks of their specific feeding habits: 0 (negligible risk), 1 (low risk), 2 (slight risk), 3 (medium risk), 4 (high risk), or 5 (extremely high risk).
		p24	(0) The introduced species have little impact on the indigenous aquatic animals in the receiving waters. (1) The introduced species may have a certain negative impact on the indigenous aquatic animals in the receiving waters. (2) The introduced species will have a certain negative impact on the indigenous aquatic animals in the receiving waters, but it can be completely controlled. (3) The introduced species will have a greater negative impact on the indigenous aquatic animals in the receiving waters, but it is still within the controllable range. (4) The introduced species will have a huge negative impact on the indigenous aquatic animals in the receiving waters. (5) The introduced species will have a very serious and irreversible negative impact on the indigenous aquatic animals in the receiving waters.
		p25	(0) The introduced species have little impact on the abiotic environment of the introduced waters. (1) The introduced species may have some negative impact on the abiotic environment of the introduced waters. (2) The introduced species will have a certain negative impact on the abiotic environment of the introduced waters, but it can be completely controlled. (3) The introduced species will have a greater negative impact on the abiotic environment of the introduced waters, but it is still within the controllable range. (4) The introduced species will have a huge negative impact on the abiotic environment of the introduced waters. (5) The introduced species will have a very serious and irreversible negative impact on the abiotic environment of the introduced waters.
		p26	(0) The introduced species hardly have a negative impact on the biodiversity of the region. (1) The introduced species have little negative impact on the biodiversity in the region. (2) The presence of introduced species pose a potential threat to biodiversity in the region. (3) The introduced species will harm the biodiversity in the region by feeding on algae or aquatic plant in large quantities, carrying harmful organisms, etc., but it is still within the controllable range. (4) The destruction of biodiversity in the region by introducing species has exceeded the self-regulation range of the ecosystem and human control range. (5) Multiple factors have led to a sharp decrease in biodiversity in the region.

**Table 3 animals-13-02035-t003:** Assessment criteria for the entry assessment.

Primary Index	Secondary Index	Tertiary Index	Assessment Criteria and Evaluation Bases
			0 (Negligible Risk), 1 (Lower Risk), 2 (Low Risk), 3 (Medium Risk), 4 (High Risk), 5 (Extremely High Risk)
R2	P3	p31	On the basis of the reference materials and expert suggestions, discuss the risks of species entry from the perspective of trade between the exporting and importing countries and assign a grade according to the risk profile: 0 (negligible risk), 1 (low risk), 2 (slight risk), 3 (medium risk), 4 (high risk), or 5 (extremely high risk).
		p32	On the basis of the reference materials and the actual situation, discuss the specific situation and responsibilities of the national fisheries and medical management organizations and assign a grade according to the risk situation: 0 (negligible risk), 1 (low risk), 2 (slight risk), 3 (medium risk), 4 (high risk), or 5 (extremely high risk).
		p33	On the basis of the actual situation, discuss the local fisheries and medical management organizations and their responsibilities in the specific export and import places of both countries and assign a grade according to the risk situation: 0 (negligible risk), 1 (low risk), 2 (slight risk), 3 (medium risk), 4 (high risk), or 5 (extremely high risk).
		p34	Discuss the aquatic animal health laws and regulations system (with emphasis on the importing country) and assign a grade according to the risk degree: 0 (negligible risk), 1 (low risk), 2 (slight risk), 3 (medium risk), 4 (high risk), or 5 (extremely high risk).
		p35	Discuss the obligations of stakeholders and assign a grade according to the risk degree: 0 (negligible risk), 1 (low risk), 2 (slight risk), 3 (medium risk), 4 (high risk), or 5 (extremely high risk).
		p36	On the basis of the reference materials and the actual situation, discuss the specific situation of the fisheries and medical diagnosis laboratory recognized by the two countries and assign a grade according to the risk degree: 0 (negligible risk), 1 (low risk), 2 (slight risk), 3 (medium risk), 4 (high risk), or 5 (extremely high risk).
		p37	On the basis of the actual health conditions of both countries, discuss the specific conditions of fishery and medical laboratories at all levels (district level, city level, and provincial level) recognized by both countries and assign a grade according to the risk degree: 0 (negligible risk), 1 (low risk), 2 (slight risk), 3 (medium risk), 4 (high risk), or 5 (extremely high risk).
		p38	On the basis of the reference materials and the actual situation, discuss the accreditation and management of the national reference laboratories of both countries and assign a grade according to the risk degree: 0 (negligible risk), 1 (low risk), 2 (slight risk), 3 (medium risk), 4 (high risk), or 5 (extremely high risk).
		p39	On the basis of the actual situation, discuss the national animal health fund support and assign a grade according to the fund support situation: 0 (negligible risk), 1 (low risk), 2 (slight risk), 3 (medium risk), 4 (high risk), or 5 (extremely high risk).
	P4	p41	(0) The introduced species hardly carry pathogens. (1) The introduced species carry a small number of pathogens of a single type. (2) The introduced species carry a single pathogen but only in small numbers. (3) The introduced species carry one or several pathogenic species, and the number is large and widely distributed. (4) The introduced species carry many kinds of pathogens at the same time, with a large number and wide distribution. (5) The introduced species carry a wide variety of pathogens, with a large number and a wide distribution.
		p42	(0) Almost no danger. (1) Extremely low hazard level with low probability of outbreak. (2) Low hazard level with slight danger. (3) Moderate hazard level and can be controlled. (4) High hazard level. (5) Extremely high hazard level and difficult to control.
		p43	Discuss the epidemic situation of diseases in the exporting countries in the past five years and assign a grade according to the risk degree: 0 (negligible risk), 1 (low risk), 2 (slight risk), 3 (medium risk), 4 (high risk), or 5 (extremely high risk).
		p44	(1) The exporting country always attaches great importance to the disease. (2) The exporting country pays sufficient attention to the disease. (3) The disease is of concern to the exporting country. (4) The exporting country pays little attention to the disease. (5) The exporting country pays little attention to the disease.
	P5	p51	Discuss the notifications of aquatic animal epidemics in both countries and assign a grade according to the risk degree: 0 (negligible risk), 1 (low risk), 2 (slight risk), 3 (medium risk), 4 (high risk), or 5 (extremely high risk).
		p52	In combination with the reference materials provided by both countries on the occurrence and disposal of the epidemic situation, summarize the occurrence and disposal of the epidemic situation and assign a grade according to the risk degree: 0 (negligible risk), 1 (low risk), 2 (slight risk), 3 (medium risk), 4 (high risk), or 5 (extremely high risk).
		p53	Summarize the management measures and actual management situation for the regionalization of specific epidemic diseases (biosafety isolation zone) in both countries and assign a grade according to the risk degree: 0 (negligible risk), 1 (low risk), 2 (slight risk), 3 (medium risk), 4 (high risk), or 5 (extremely high risk).
		p54	Summarize the national animal epidemic monitoring plan and actual implementation situation of both countries and assign a grade according to the risk degree: 0 (negligible risk), 1 (low risk), 2 (slight risk), 3 (medium risk), 4 (high risk), or 5 (extremely high risk).
		p55	Summarize the vector monitoring plan and actual implementation situation of both countries and assign a grade according to the risk degree: 0 (negligible risk), 1 (low risk), 2 (slight risk), 3 (medium risk), 4 (high risk), or 5 (extremely high risk).
		p56	Discuss the epidemic immunity of the aquatic animals to be introduced and assign a grade according to the risk degree: 0 (negligible risk), 1 (low risk), 2 (slight risk), 3 (medium risk), 4 (high risk), or 5 (extremely high risk).
		p57	Summarize the identification situation and traceability of aquatic animals to be introduced and assign a grade according to the risk degree: 0 (negligible risk), 1 (low risk), 2 (slight risk), 3 (medium risk), 4 (high risk), or 5 (extremely high risk).
		p58	According to the relevant management requirements for the registration of aquaculture farms and biosafety in both countries, assign a grade according to the risk degree: 0 (negligible risk), 1 (low risk), 2 (slight risk), 3 (medium risk), 4 (high risk), or 5 (extremely high risk).
		p59	Discuss the coastal port quarantine measures of both countries and assign a grade according to the risk degree: 0 (negligible risk), 1 (low risk), 2 (slight risk), 3 (medium risk), 4 (high risk), or 5 (extremely high risk).
	P6	p61	(1) The introduction will be small in quantity and scale. (2) The number and scale of introduced species will be small (larger than (1)). (3) The introduction will have a certain number and scale. (4) The number and scale of the introduction will be large. (5) The introduction will be carried out on a huge scale.
		p62	Since there is no direct linear relationship between the economic benefits and risks generated by the introduced species, users can generalize the specific situation of the economic benefits generated by the introduced species. Explain the possible relationship between the generated economic benefits and the risks and assign a grade according to the risk situation: 0 (negligible risk), 1 (low risk), 2 (slight risk), 3 (medium risk), 4 (high risk), or 5 (extremely high risk).
		p63	(1) The costs of isolation and quarantine are very low. (2) The costs of isolation and quarantine are low. (3) The costs of isolation and quarantine are within the acceptable range. (4) The costs of isolation and quarantine are high. (5) The costs of isolation and quarantine are extremely high.
		p64	(1) The costs of each intermediate link are very low. (2) The costs of each intermediate link are low. (3) The costs of each intermediate link are within the acceptable range. (4) The costs of each intermediate link are high. (5) The costs of each intermediate link are extremely high.

**Table 4 animals-13-02035-t004:** Assessment criteria for the exposure assessment.

Primary Index	Secondary Index	Tertiary Index	Assessment Criteria and Evaluation Bases
			0 (Negligible Risk), 1 (Lower Risk), 2 (Low Risk), 3 (Medium Risk), 4 (High Risk), 5 (Extremely High Risk)
R3	P7	p71	List the possible pathogens that may exist during the introduction process based on the reference literature and expert opinions and assign a grade of risk according to their biological characteristics (pathogen type, basic characteristics, virulence, etc.): 0 (negligible risk), 1 (low risk), 2 (slight risk), 3 (medium risk), 4 (high risk), or 5 (extremely high risk).
		p72	(0) There is one host species, and the number is very small. (1) There is one host species, and the number is small. (2) There is one host species, but it has an appreciable number. (3) There is a single host species, but the number is large, and it is widely distributed. (4) There are multiple host species, with a large number and wide distribution. (5) There are a variety of host species, with a large number and a wide distribution.
		p73	(0) The infection rate is almost zero. (1) Infections occur from time to time but cannot cause disease. (2) There is a certain risk of infection, but the risks of morbidity and mortality are low. (3) The infection situation is relatively widespread, with a certain risk of morbidity and mortality. (4) The infection is widespread and can cause an epidemic accompanied by a certain degree of mortality. (5) The emergence of pathogens is accompanied by extremely high infection, morbidity, and mortality rates.
		p74	Combined with the relevant data, discuss the threat of pathogens to human beings and assign a grade according to the risk degree: 0 (negligible risk), 1 (low risk), 2 (slight risk), 3 (medium risk), 4 (high risk), or 5 (extremely high risk).
	P8	p81	(0) Pathogens have almost no transmission medium or route. (1) The transmission vector and route of the pathogens are very minor. (2) Pathogens have vectors and routes of transmission but have little impact on species. (3) Pathogens have one or several transmission media and transmission routes. (4) Pathogens have many kinds of transmission media and routes. (5) Pathogenic vectors and transmission routes are extremely complex and diverse.
		p82	(0) Pathogens can hardly spread. (1) Pathogens’ transmission speed and capacity do not pose a threat to species (indigenous species and introduced species). (2) Pathogens’ transmission speed is low, and their transmission ability is weak. (3) Pathogens’ transmission speed is considerable, with a certain transmission capacity. (4) Pathogens’ transmission speed is fast, with a strong transmission ability, which poses a certain threat to species. (5) Pathogens spread very fast and have strong transmission ability, which poses a huge threat to species.
		p83	(1) Nonbiological factors have inhibitory effects on the spread of pathogens and are far greater than promoting effects. (2) Nonbiological factors both inhibit and promote the spread of pathogens (promotion < inhibition). (3) Nonbiological factors both promote and inhibit the spread of pathogens (promotion > inhibition). (4) Nonbiological factors have a definite promotion effect on the spread of pathogens and are far greater than the inhibition effect. (5) Nonbiological factors can greatly promote the spread of pathogens.
	P9	p91	Combined with the literature related to the pathogen and the actual infection situation of diseased hosts, provide a summary of the host situation and infection symptoms and assign a grade of risk degree: 0 (negligible risk), 1 (low risk), 2 (slight risk), 3 (medium risk), 4 (high risk), or 5 (extremely high risk).
		p92	On the basis of the literature related to the pathogen and the actual infection situation of diseased hosts, discuss the infection site of the pathogen and its specific infection situation and assign a grade of risk degree: 0 (negligible risk), 1 (low risk), 2 (slight risk), 3 (medium risk), 4 (high risk), or 5 (extremely high risk).
		p93	On the basis of the literature related to the pathogen and the actual infection situation of diseased hosts, discuss the pathological changes of pathogenic infection and assign a grade of risk degree: 0 (negligible risk), 1 (low risk), 2 (slight risk), 3 (medium risk), 4 (high risk), or 5 (extremely high risk).
		p94	Combined with the literature related to the pathogen and the actual infection situation of diseased hosts, discuss the specific situation (age, sex, etc.) of diseased hosts and assign a grade according to the risk degree: 0 (negligible risk), 1 (low risk), 2 (slight risk), 3 (medium risk), 4 (high risk), or 5 (extremely high risk).
	P10	p101	Combined with the relevant data on vaccine use, discuss the status of vaccines against the above pathogens (including vaccine type, use method, use cost, vaccine potency, and other factors) and assign a grade according to the risk degree: 0 (negligible risk), 1 (low risk), 2 (slight risk), 3 (medium risk), 4 (high risk), or 5 (extremely high risk).
		p102	Combined with the reference literature and the actual pathogen detection situation, discuss the detection methods of the above pathogens (including the detection method type, detection cost and effect, and other factors) and assign a grade according to the risk degree: 0 (negligible risk), 1 (low risk), 2 (slight risk), 3 (medium risk), 4 (high risk), or 5 (extremely high risk).
		p103	Referring to the literature of the fishery medical diagnosis laboratory and the pathogen prevention measures in actual breeding, discuss the prevention measures against the above pathogens and assign a grade according to the risk degree: 0 (negligible risk), 1 (low risk), 2 (slight risk), 3 (medium risk), 4 (high risk), or 5 (extremely high risk).
		p104	Referring to the literature of the fishery medical diagnosis laboratory and the pathogen treatment methods in actual breeding, discuss the treatment methods for the above pathogens and assign a grade according to the risk degree: 0 (negligible risk), 1 (low risk), 2 (slight risk), 3 (medium risk), 4 (high risk), or 5 (extremely high risk).

**Table 5 animals-13-02035-t005:** Assessment criteria for the consequence assessment.

Primary Index	Secondary Index	Tertiary Index	Assessment Criteria and Evaluation Bases
			0 (Negligible Risk), 1 (Lower Risk), 2 (Low Risk), 3 (Medium Risk), 4 (High Risk), 5 (Extremely High Risk)
R4	P11	p111	(0) The infection rate is almost zero. (1) The infection rate is very low. (2) The infection rate is low. (3) There is an appreciable infection rate, but it is within the controllable range. (4) The infection rate is high. (5) The infection rate is extremely high.
		p112	(0) The incidence rate is almost zero. (1) The incidence rate is very low. (2) The incidence rate is low. (3) There is an appreciable incidence rate, but it is within the controllable range. (4) The incidence rate is high. (5) The incidence rate is extremely high.
		p113	(0) The mortality rate is almost zero. (1) The mortality rate is very low. (2) The mortality rate is low. (3) There is an appreciable mortality rate but no uncontrollable impact. (4) The mortality rate is high. (5) The mortality rate is extremely high.
		p114	(0) There is almost no adverse effect on human health. (1) Introduction may cause a decline in the human immune function and pose a potential threat to human health. (2) Introduction will have adverse effects on human health, but it will not cause disease. (3) Pathogens may cause diseases and pose a threat to human health. (4) There have been cases in history of endangering human health. (5) There have been cases of individual or group harm or death.
		p115	(0) There is almost no impact on the economy. (1) Introduction will cause direct economic losses but have little impact. (2) Direct economic losses will be incurred, but all of it will be within the controllable range. (3) Introduction will cause direct economic losses and is difficult to control. (4) The direct economic losses will be large and very difficult to control. (5) Introduction will cause enormous damage to the economy and immeasurable economic losses.
	P12	p121	(1) The costs of monitoring, prevention, and control are very low, and the effect is significant. (2) The costs of monitoring, prevention, and control are low, and the effect is remarkable. (3) The costs of monitoring, prevention, and control are high, and the effect is moderate. (4) The costs of monitoring, prevention, and control are high, and the effect is poor. (5) There are no effective prevention and control methods, or the extremely high costs render the measures almost infeasible.
		p122	(0) There are almost no potential transaction losses. (1) The potential transaction losses are very low. (2) The potential transaction losses are slight. (3) The potential transaction losses generated are within the controllable range. (4) The potential transaction losses are large and difficult to control. (5) The potential transaction losses generated are huge and beyond the scope of control.
		p123	(0) There is almost no potential impact on the social economy. (1) Introduction has potential impacts on the social economy, but it has little impact. (2) The potential social economic impacts are under control. (3) The potential social economic impacts are conspicuous and cannot be ignored. (4) The potential social economic impacts are huge and difficult to control. (5) Introduction will produce immeasurable social economic potential impacts and will cause very serious economic losses.
		p124	(0) There is basically no impact on the ecological environment in the region. (1) The impact on the ecological environment in the region is slight. (2) The impact on the ecological environment in the region is basically controllable. (3) Introduction will have an appreciable impact on the ecological environment in the region, but it can be restored through ecosystem regulation. (4) Introduction will cause an appreciable degree of damage to the ecological environment in the region. (5) Introduction will have a sustained, serious, and far-reaching impact on the ecological environment in the region.
		p125	(0) There is almost no damage to biodiversity. (1) Introduction will hurt the biodiversity of the region. (2) Introduction will cause some damage to the biodiversity of the region. (3) Introduction will cause harm to the biodiversity of the region, but it can be controlled. (4) Introduction will destroy the biodiversity of the region, and it will be difficult to control. (5) Introduction will cause irreparable serious damage to the biodiversity of the region.
		p126	(1) The restoration costs are very low, and the effect is significant. (2) The restoration costs are low, and the effect is remarkable. (3) The restoration costs are high, and the effect is moderate. (4) The restoration costs are high, and the effect is poor. (5) The restoration costs are huge and difficult to achieve.

**Table 6 animals-13-02035-t006:** Specific situations and scoring evaluation rules for indices to be discussed.

Score	Specific Situation and Scoring Evaluation Rules for Indices to Be Discussed
0	(1) Inability to generate risks or minimal likelihood of generating risks (negligible); (2) The risk cannot generate any harm or loss; (3) There are practical and effective ways to eliminate the risks.
1	(1) Low likelihood of risk generation; (2) External factors prevent the sustained existence of the risks; (3) There are effective measures and means to prevent and control the risks before they have a negative impact.
2	(1) There is a possibility of generating risks, but the harm, losses, or negative impacts brought by the risks are relatively small; (2) Relevant policies and measures can reduce the harm and losses caused the by risks to an acceptable range; (3) There are various practical and effective means to eliminate the negative impact of the risks.
3	(1) There is a possibility of generating risks, but the hazards, losses, or negative effects of the risks are within a controllable range; (2) Relevant policies and measures can provide guarantees for risk prevention and control; (3) There are relatively feasible and effective means to control the negative impact of the risks.
4	(1) The likelihood of risk generation is high, and the harm, losses, or negative impacts of risk generation are difficult to control; (2) The harm, losses, or negative impacts generated by the risks have an appreciable degree of sustainability; (3) External factors make it difficult or ineffective to implement relevant prevention and control measures.
5	(1) The harm, losses, or negative impacts caused by the risks are significant and irreparable; (2) The complex and diverse mechanisms of risk generation and diffusion lead to a huge scope of harm and strong sustainability; (3) There is no effective method for preventing and controlling the risks; (4) Because of policy loopholes and high costs, some intermediate links during the introduction process cannot be implemented; thus, protection for the introduction cannot be provided.

**Table 7 animals-13-02035-t007:** Scoring of tertiary indices of introduced aquatic animals.

Tertiary Index	*Pterygoplichthys pardalis*	*Macrobrachium rosenbergii*	*Crassostrea gigas*	*Trachemys scripta elegans*	*Ambystoma mexicanum*
p11	5	2	2	4	0
p12	0	0	0	0	1
p21	5	4	5	5	3
p22	4	4	4	5	3
p23	5	3	3	5	3
p24	4	4	3	5	1
p25	5	3	3	4	1
p26	4	4	3	5	1
p31	2	2	3	3	3
p32	1	1	1	1	1
p33	1	1	1	1	1
p34	1	1	1	1	1
p35	1	1	1	1	1
p36	1	1	1	1	1
p37	1	1	1	1	1
p38	2	2	1	1	2
p39	2	2	2	2	2
p41	4	4	3	2	3
p42	5	5	5	3	4
p43	1	1	1	1	1
p44	1	1	1	2	1
p51	1	1	1	1	1
p52	1	1	1	1	1
p53	1	1	1	1	1
p54	1	1	1	1	1
p55	1	1	1	1	1
p56	3	4	4	2	3
p57	2	2	2	2	2
p58	2	2	2	2	2
p59	1	1	1	1	1
p61	3	5	4	3	2
p62	4	4	3	2	3
p63	3	3	2	2	4
p64	3	3	2	2	4
p71	5	5	4	2	3
p72	5	5	4	3	4
p73	4	5	5	3	4
p74	2	2	2	1	1
p81	4	5	3	3	4
p82	5	5	4	3	4
p83	4	4	2	3	3
p91	5	5	4	3	4
p92	5	5	4	3	4
p93	4	5	4	2	3
p94	5	5	5	4	4
p101	5	5	5	4	4
p102	3	1	2	1	1
p103	4	1	2	1	2
p104	4	4	4	3	2
p111	5	5	5	3	3
p112	4	5	4	2	3
P113	4	5	4	1	3
P114	2	2	2	1	2
P115	5	5	4	3	4
P121	5	4	3	4	4
P122	4	4	4	3	3
P123	4	4	3	3	2
P124	5	4	2	4	3
P125	5	3	2	5	2
P126	5	3	2	4	3

**Table 8 animals-13-02035-t008:** Calculation results for introduced aquatic animals.

Index	Introduced Aquatic Animals				
	*Pterygoplichthys pardalis*	*Macrobrachium rosenbergii*	*Crassostrea gigas*	*Trachemys scripta elegans*	*Ambystoma mexicanum*
P1	3.3335	1.3334	1.3334	2.6668	0.3333
P2	4.3843	3.7023	3.3882	4.7456	1.6898
P3	1.2362	1.2362	1.2205	1.2205	1.2811
P4	3.9472	3.9472	3.6984	2.4676	3.1482
P5	1.2295	1.2965	1.2965	1.1625	1.2295
P6	3.4596	3.7740	2.7740	2.1572	3.2260
P7	3.7377	3.9314	3.4813	1.9626	2.6064
P8	4.5907	4.9245	3.5152	3.0000	3.9245
P9	4.6663	5.0000	4.0499	2.7162	3.6663
P10	3.8854	2.1522	2.8230	1.8230	2.0500
P11	3.7002	3.9770	3.3782	1.8645	2.9810
P12	4.6164	3.8380	2.9590	3.6722	3.1360
R1	3.6837	2.1230	2.0183	3.3597	0.7854
R2	2.8924	2.9601	2.6812	1.9539	2.4741
R3	3.9908	3.4341	3.2683	2.1084	2.6469
R4	3.8843	3.9492	3.2944	2.2260	3.0120
R	3.6973	2.9622	2.7073	2.5948	1.9742
Risk grade	high	medium	medium	medium	low

## Data Availability

All data will be made available upon request from the corresponding author.

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
