# Peer review of "Risk Assessment Model System for Aquatic Animal Introduction Based on Analytic Hierarchy Process (AHP)"

_animals, 2023, doi:10.3390/ani13122035_

Round 1
Reviewer 1 Report
Dear Authors,
I found this research interesting and, above all, of effective utility for the management and conservation of marine biodiversity. The evaluation of the risks related to biological invasions is of great relevance for the ecosystems. Methods is good, and results are exposed in a clear manner. However, in the introduction parte a pair of sentences must be added about the opening of canals such as, for example, the Suez Canal and its implication for Mediterranean biodiversity, economy and human health for humans. Please, at this regard, quote Tiralongo et al., 2020 (Snapshot of rare, exotic and overlooked fish species in the Italian seas: A citizen science survey). Published in Journal of Sea Research.
I suggest only a general check for some minor changes.
Author Response
I found this research interesting and, above all, of effective utility for the management and conservation of marine biodiversity. The evaluation of the risks related to biological invasions is of great relevance for the ecosystems. Methods is good, and results are exposed in a clear manner. However, in the introduction part a pair of sentences must be added about the opening of canals such as, for example, the Suez Canal and its implication for Mediterranean biodiversity, economy and human health for humans. Please, at this regard, quote Tiralongo et al., 2020 (Snapshot of rare, exotic and overlooked fish species in the Italian seas: A citizen science survey). Published in Journal of Sea Research.
Response: Thank you for your suggestion. We believe that the literature you provided has a great benefit to our research. We have added this literature in the introduction part as a reference.
Line 122 to 127:
At the same time, these changes are constantly affecting human activities such as ship trade and transport, working together with some complex natural factors, resulting in a larger range of species changes. For example, the opening of the Suez Canal has significantly changed the diversity of fish in the Mediterranean region, which has had an impact on Italy's socio-economic and human health [34].
[34] Tiralongo, F.; Crocetta, F.; Riginella, E.; Lillo, A.O.; Tondo, E.; Macali, A.; Mancini, E.; Russo, F.; Coco, S.; Paolillo, G.; et al. Snapshot of rare, exotic and overlooked fish species in the Italian seas: A citizen science survey. Journal of Sea Research 2020, 164, 101930.
Reviewer 2 Report
General
The authors present a method to assess the risk of alien species introduced for animal husbandry to become invasive. The method, based on the Analytic Hierarchy Process, is based on a comprehensive list of criteria that include autoecological information, but also other societal criteria such as the quality of the medical systems in the exporting and importing countries. In this sense, it seems to be a step forward in the extremely difficult task of forecasting and preventing the spread of invasive species. On the wrong side, for many potentially invasive species, experts will simply have no enough information to rate many of the criteria, what could lead to subjective and inconsistent results. But I imagine even this is better than nothing or better than the more simple assessment tools usually used.
The manuscript is overall well written, although some small changes (listed below) can make it clearer for the reader. The literature cited is complete and well updated. Therefore, the paper could be published in Animals after minor revision.
Specific comments
A first comment on terminology. The authors include the term “germoplasm resource” in several places of the manuscript, including the title. This term adds nothing but complexity and would be better avoided. For instance, the title would have the same meaning but read much better as: “A Risk Assessment Procedure for Aquatic Animal Introduction Based on Analytic Hierarchy Process”, or “An analytic hierarchy process method to assess the risk of invasion by aquaculture animals”. Better remove all references to germoplasm resource through the manuscript.
Also, the use of the term “agriculture” through the text is unfortunate, as the authors mean it to encompass forestry and animal husbandry, whereas most readers would understand agriculture as an activity totally different from forestry and husbandry. For the current paper, the term “aquaculture” would be much more suited.
A minor terminological issue: the plural of index is indices, not indexes. Substitute throughout the text.
In addition to terminology, the procedure itself of the model is not totally clear. Table 1, for instance, presents the primary, secondary and tertiary indices used to assess the risk of invasion, in what is a very comprehensive list. But the table heading should explain it in greater detail. For instance, the tertiary indices, they all have the same range? I mean, p11 and p12 can go from 0 to 1, depending on the species? Or does each tertiary index have a different range? If the latter is the case, does not this pose a problem? Wouldn’t it better to have all indices with the same range and then to weigh each index differently according to its relevance for each general criterion?
Other issues.
P1. What is “relative typical” introduced animals? Explain.
P1. What is scientific introduction of animals? Are scientists introducing animals? Do you mean importing to a country or introducing into the wild? Please, rephrase.
P2. What is the “widespread scope” of an animal? Rephrase.
P2 paragraph 1. The authors give two definitions that contrast in the consideration of “introduced”. Do invasives need to be introduced, or can species that arrived by themselves be also considered invasive? Please, make clear what is the definition you use.
P2. “Developing more scientific introduction strategies”? Do we really need more introductions? What for? The authors should justify this.
P3. “After more than 20 years of research and development”? Better 40 years?
P3 paragraph 1. Adjust format of references to the standard of the journal. For instance, Mehmet Cihan Aydin et al. (2022).
P3 last paragraph. What epidemic diseases are you considering? Only those affecting man, or also those on other wild or domestic species? Please, make clear.
P4. Here authors mention impacts on human health. Invasives often are vectors of disease for native species other than humans. Crayfish are a good example. Please, make it clear whether you take this into account or not
Table 1, p25. Introduced waters? You are not introducing water, are you? I think you mean receiving waters.
Table 1, p26. Isn’t this recurrent with p24? Explain differences.
Table 1, p37. Unclear what “all levels” are.
Page 5, section 2.2. Explain what were the experts in your case. How many, how were they selected, and how can this selection bias the results.
Page 14. What field observations? Explain which field observations were performed in your case.
Page 16. “Pterygoplichthys pardalis has now established large natural populations”. Natural populations? Do you mean wild populations?
Pae 19. What are “their bodies or reproductive locales”?
The quality of English language is good, but need some minor changes, explained in detail above
Author Response
Response to Reviewer 2:
1. A first comment on terminology. The authors include the term “germplasm resource” in several places of the manuscript, including the title. This term adds nothing but complexity and would be better avoided. For instance, the title would have the same meaning but read much better as: “A Risk Assessment Procedure for Aquatic Animal Introduction Based on Analytic Hierarchy Process”, or “An analytic hierarchy process method to assess the risk of invasion by aquaculture animals”. Better remove all references to germoplasm resource through the manuscript.
Response: Thank you for your suggestion. We have removed all descriptions about “germplasm resource” throughout the manuscript and changed the title to "Risk Assessment Model System for Aquatic Animal Introduction Based on Analytic Hierarchy Process (AHP)".
2. Also, the use of the term "agriculture" through the text is unfortunate, as the authors mean it to encompass forestry and animal husbandry, whereas most readers would understand agriculture as an activity totally different from forestry and husbandry. For the current paper, the term “aquaculture” would be much more suited.
Response: Thank you for your suggestion. We have changed the "agriculture" in the manuscript to more appropriate terms such as "aquaculture", "animal husbandry" or "animal production".
Line 106 to 107:
Currently, AHP is widely used in animal production, resource utilization, natural disaster prevention and control, and other fields.
Line 115 to 119: The aquatic ecosystem is an important part of the earth's environment, and a healthy aquatic ecological environment is an important prerequisite not only for aquatic ecosystems to provide services and perform their functions but also to present a solid guarantee for the sustainable and healthy development of modern aquaculture and animal husbandry.
3. A minor terminological issue: the plural of index is indices, not indexes. Substitute throughout the text.
Response: Thank you for your suggestion. We have changed "indexes" to "indices" throughout the MS.
4. In addition to terminology, the procedure itself of the model is not totally clear. Table 1, for instance, presents the primary, secondary and tertiary indices used to assess the risk of invasion, in what is a very comprehensive list. But the table heading should explain it in greater detail. For instance, the tertiary indices, they all have the same range? I mean, p11 and p12 can go from 0 to 1, depending on the species? Or does each tertiary index have a different range? If the latter is the case, does not this pose a problem? Wouldn’t it better to have all indices with the same range and then to weigh each index differently according to its relevance for each general criterion?
Response: Thank you for your suggestion. We believe that having a unified assignment range can indeed make the model clearer. Here, we have made modifications to the content and assignment criteria of the p11 and p12 indices in Table 1 and Table 2, as well as scoring results of the tertiary index in Table 7, calculation results of the secondary index and primary index in Table 8, and the discussion content in the appendix tables.
Table 1. The indices of the risk assessment system for the introduction of aquatic animals.
|
Primary Index |
Secondary Index |
Tertiary Index |
|
Hazard assessment of introduced species (R1) |
Basic attributes of introduced species (P1) |
Basic invasive situation of non-native species (p11) |
|
Basic endangered situation of non-native species (p12) |
||
|
Self-hazard of introduced species (P2) |
Suitability of environmental factors (including temperature, dissolved oxygen, salinity, pH, etc.) (p21) |
|
|
Natural enemies of introduced species (p22) |
||
|
Feeding habits of introduced species (p23) |
||
|
Impact of introduced species on indigenous aquatic animals in receiving waters (p24) |
||
|
Impact of introduced species on the abiotic environment of the receiving waters (p25) |
||
|
Impact of introduced species on biodiversity (mainly referring to the impact on algae, aquatic plants, microorganisms, etc.) (p26) |
Table 2. Assessment criteria for the hazard assessment of introduced species.
|
Primary Index |
Secondary Index |
Tertiary Index |
Assessment Criteria and Evaluation Bases |
|
0 (Negligible Risk), 1 (Low Risk), 2 (Slight Risk), 3 (Medium Risk), 4 (High Risk), 5 (Extremely High Risk) |
|||
|
R1 |
P1 |
p11 |
Based on the list of invasive species in the importing country and the actual situation of non-native species, make a basic judgment on the invasion situation of the introduced species and assign a grade according to the invasion risk of introduced species: 0 (negligible risk), 1 (low risk), 2 (slight risk), 3 (medium risk), 4 (high risk), or 5 (extremely high risk). |
|
p12 |
Based on the IUCN Red List of Threatened Species and the actual situation of the importing country, make a basic judgment on the endangered situation of the introduced species and assign a grade according to the risk of introduced species: 0 (negligible risk), 1 (low risk), 2 (slight risk), 3 (medium risk), 4 (high risk), or 5 (extremely high risk). |
Table 7. Scoring of tertiary indices indexes of introduced aquatic animals.
|
Tertiary Index |
Pterygoplichthys pardalis |
Macrobrachium rosenbergii |
Crassostrea gigas |
Trachemys scripta elegans |
Ambystoma mexicanum |
|
p11 |
5 |
2 |
2 |
4 |
0 |
|
p12 |
0 |
0 |
0 |
0 |
1 |
Table 8. Calculation results for introduced aquatic animals.
|
Index |
Introduced Aquatic Animals |
||||
|
Pterygoplichthys pardalis |
Macrobrachium rosenbergii |
Crassostrea gigas |
Trachemys scripta elegans |
Ambystoma mexicanum |
|
|
P1 |
3.3335 |
1.3334 |
1.3334 |
2.6668 |
0.3333 |
|
P2 |
4.3843 |
3.7023 |
3.3882 |
4.7456 |
1.6898 |
|
P3 |
1.2362 |
1.2362 |
1.2205 |
1.2205 |
1.2811 |
|
P4 |
3.9472 |
3.9472 |
3.6984 |
2.4676 |
3.1482 |
|
P5 |
1.2295 |
1.2965 |
1.2965 |
1.1625 |
1.2295 |
|
P6 |
3.4596 |
3.7740 |
2.7740 |
2.1572 |
3.2260 |
|
P7 |
3.7377 |
3.9314 |
3.4813 |
1.9626 |
2.6064 |
|
P8 |
4.5907 |
4.9245 |
3.5152 |
3.0000 |
3.9245 |
|
P9 |
4.6663 |
5.0000 |
4.0499 |
2.7162 |
3.6663 |
|
P10 |
3.8854 |
2.1522 |
2.8230 |
1.8230 |
2.0500 |
|
P11 |
3.7002 |
3.9770 |
3.3782 |
1.8645 |
2.9810 |
|
P12 |
4.6164 |
3.8380 |
2.9590 |
3.6722 |
3.1360 |
|
R1 |
3.6837 |
2.1230 |
2.0183 |
3.3597 |
0.7854 |
|
R2 |
2.8924 |
2.9601 |
2.6812 |
1.9539 |
2.4741 |
|
R3 |
3.9908 |
3.4341 |
3.2683 |
2.1084 |
2.6469 |
|
R4 |
3.8843 |
3.9492 |
3.2944 |
2.2260 |
3.0120 |
|
R |
3.6973 |
2.9622 |
2.7073 |
2.5948 |
1.9742 |
|
Risk grade |
high |
medium |
medium |
medium |
low |
Table A1. Pterygoplichthys pardalis.
|
Indexes |
Content of the Discussion |
|
p11 |
Pterygoplichthys pardalis has been explicitly listed as an invasive species by the Chinese government. Therefore, the risk of biological invasion is extremely high. |
Table A2. Macrobrachium rosenbergii.
|
Indexes |
Content of the Discussion |
|
p11 |
Macrobrachium rosenbergii is an economic shrimp, not an invasive non-native species. However, large-scale cultivation of Macrobrachium rosenbergii also has a certain risk of biological invasion. |
Table A3. Crassostrea gigas.
|
Indexes |
Content of the Discussion |
|
p11 |
Crassostrea gigas is an economic shellfish, not an invasive non-native species. However, large-scale cultivation of Crassostrea gigas also has a certain risk of biological invasion. |
Table A4. Trachemys scripta elegans.
|
Indexes |
Content of the Discussion |
|
p11 |
Trachemys scripta elegans has been listed by IUCN as one of the 100 most destructive invasive species. Therefore, the risk of biological invasion is high. |
Table A5. Ambystoma mexicanum.
|
Indexes |
Content of the Discussion |
|
p11 |
Ambystoma mexicanum are sold mainly as pets, and they are not an invasive non-native species. |
|
p12 |
Currently, Ambystoma mexicanum has been included in the IUCN Red List of Threatened Species (Critically Endangered, CR), but artificial cultivation has been achieved in China, so the risk of Ambystoma mexicanum is relatively low. |
5. What is “relative typical” introduced animals? Explain.
Response: Thank you. When selecting the assessment object, we try our best to include fish, crustaceans, shellfish, and amphibians and also cover different introduction purposes, such as animal food production, ornamental aquatic animal, and genetic breeding. Therefore, relatively typical only wants to express that these species have a certain degree of representativeness. But to make it easier to understand, we have changed the expression of "relatively typical" to more straightforward descriptions.
Line 34 to 39:
To fill this gap, we used an analytic hierarchy process (AHP) to build a risk assessment model system for aquatic IS. Our AHP has four primary indexes, twelve secondary indexes, and sixty tertiary indexes. We used this AHP to conduct quantitative risk assessments on five aquatic animals that are typically introduced in China, which have distinct biological characteristics and specific introduction purposes, and can represent different types of aquatic animals.
6. What is scientific introduction of animals? Are scientists introducing animals? Do you mean importing to a country or introducing into the wild? Please, rephrase.
Response: Thank you for your suggestion. “Scientific” here refers to the safe and reasonable introduction of non-native species into the importing country. We have changed the original sentence for a better understanding.
Line 41 to 44:
Risk assessment of introduction of aquatic animals via our AHP is effective and it provided support for the introduction and healthy breeding of aquatic animal. Thus, the AHP model can provide a basis for decision making risk management concerning the introduction of species.
7. What is the “widespread scope” of an animal? Rephrase.
Response: Thank you for your suggestion. We have changed “widespread scope” to “spread capability”.
Line 55 to 59:
An IS is a non-native species that has invasive biological characteristics (such as strong adaptability and spread capability) or may have negative impacts on ecology, agriculture, fisheries, food, or human health, including artificially introduced and unintentionally arrived non-indigenous species.
8. P2 paragraph 1. The authors give two definitions that contrast in the consideration of “introduced”. Do invasives need to be introduced, or can species that arrived by themselves be also considered invasive? Please, make clear what is the definition you use.
Response: Thank you for your suggestion. We overlooked the consideration of the origin of non-native species when providing the definition, and we have made the following modifications:
Line 55 to 59:
An IS is a non-native species that has invasive biological characteristics (such as strong adaptability and spread capability) or may have negative impacts on ecology, agriculture, fisheries, food, or human health, including artificially introduced and unintentionally arrived non-indigenous species.
9. “Developing more scientific introduction strategies”? Do we really need more introductions? What for? The authors should justify this.
Response: Thank you for your suggestion. The term “more” here specifically refers to an increase in degree rather than quantity. To avoid readers’ misunderstanding, we deleted “more”.
Line 88 to 92:
By conducting risk assessments on the entry of, exposure to, and consequences of introduced species and developing scientific introduction strategies based on the assessment results, the harm caused by invasive species can be reduced effectively.
10. “After more than 20 years of research and development”? Better 40 years?
Response: Thank you for your suggestion. We have changed “20” to “40”.
Line 103 to 106:
After more than 40 years of research and development, AHP has become one of the most mainstream multiple-criteria decision-making (MCDM) analysis methods, and its main advantages include simplicity, flexibility, and rigorous and strong operability.
11. P3 paragraph 1. Adjust format of references to the standard of the journal. For instance, Mehmet Cihan Aydin et al. (2022).
Response: Thank you for your suggestion. We have made the following modifications:
Line 107 to 112:
Currently, AHP is widely used in animal production, resource utilization, natural disaster prevention and control, and other fields; Hadi Veisi proposed the application of the AHP in a multi-criteria selection of agricultural irrigation systems [24]; Priyanka Yadav established a decision support system for the selection of biogas upgrade technologies based on AHP [26]; and Mehmet Cihan Aydin combined geographic information systems with AHP to assess the flood risk in Bitlis Province, Turkey [27].
Reference:
[25] Veisi, H.; Deihimfard, R.; Shahmohammadi, A.; Hydarzadeh, Y. Application of the analytic hierarchy process (AHP) in a multi-criteria selection of agricultural irrigation systems. Agric. Water Manag. 2022, 267, 107619.
[26] Yadav, P.; Yadav, S.; Singh, D.; Kapoor, R.M.; Giri, B.S. An analytical hierarchy process based decision support system for the selection of biogas up-gradation technologies. Chemosphere 2022, 302, 134741.
[27] Aydin, M.C.; Sevgi BirincioÄŸlu, E. Flood risk analysis using gis-based analytical hierarchy process: A case study of Bitlis Province. Appl. Water Sci. 2022, 12, 122.
12. P3 last paragraph. What epidemic diseases are you considering? Only those affecting man, or also those on other wild or domestic species? Please, make clear.
Response: Thank you for your suggestion. We have considered the epidemic diseases of wild and domestic aquatic animals and zoonotic diseases of aquatic animals. And we have made the following modifications:
Line 151 to 154:
If we want to carry out a specific quantitative risk assessment of an introduced non-native species, we must consider both the biological risk of the species itself and the risk of epidemic diseases (including epidemic diseases of wild and domestic aquatic animals and zoonotic diseases of aquatic animals).
13. Here authors mention impacts on human health. Invasives often are vectors of disease for native species other than humans. Crayfish are a good example. Please, make it clear whether you take this into account or not.
Response: Thank you for your suggestion. We have considered the harm to indigenous species (including becoming direct victims and pathogen vectors).
Line 158 to 162:
The risk of infectious diseases should include the risk assessment of pathogen exposure; hazards to indigenous species (including becoming direct victims and pathogen vectors); the consequences of disease outbreak; the impacts on human health, social economy, and ecological environment; and other relevant factors.
14. Table 1, p25. Introduced waters? You are not introducing water, are you? I think you mean receiving waters.
Response: Thank you for your suggestion. You are absolutely right. It should be receiving waters, and we changed them in Table 1 and Table 2.
Table 1. The indices of the risk assessment system for the introduction of aquatic animals.
|
Primary Index |
Secondary Index |
Tertiary Index |
|
Hazard assessment of introduced species (R1) |
Self-hazard of introduced species (P2) |
Suitability of environmental factors (including temperature, dissolved oxygen, salinity, pH, etc.) (p21) |
|
Natural enemies of introduced species (p22) |
||
|
Feeding habits of introduced species (p23) |
||
|
Impact of introduced species on indigenous aquatic animals in receiving waters (p24) |
||
|
Impact of introduced species on the abiotic environment of the receiving waters (p25) |
||
|
Impact of introduced species on biodiversity (mainly referring to the impact on algae, aquatic plants, microorganisms, etc.) (p26) |
Table 2. Assessment criteria for the hazard assessment of introduced species.
|
Primary Index |
Secondary Index |
Tertiary Index |
Assessment Criteria and Evaluation Bases |
|
0 (Negligible Risk), 1 (Low Risk), 2 (Slight Risk), 3 (Medium Risk), 4 (High Risk), 5 (Extremely High Risk) |
|||
|
R1 |
P2 |
p24 |
(0) The introduced species have little impact on the indigenous aquatic animals in the receiving waters. (1) The introduced species may have a certain negative impact on the indigenous aquatic animals in the receiving waters. (2) The introduced species will have a certain negative impact on the indigenous aquatic animals in the receiving waters, but it can be completely controlled. (3) The introduced species will have a greater negative impact on the indigenous aquatic animals in the receiving waters, but it is still within the controllable range. (4) The introduced species will have a huge negative impact on the indigenous aquatic animals in the receiving waters. (5) The introduced species will have a very serious and irreversible negative impact on the indigenous aquatic animals in the receiving waters. |
15. Table 1, p26. Isn’t this recurrent with p24? Explain differences.
Response: Thank you for your suggestion, p24 focuses on evaluating the direct impact of non-native species on indigenous aquatic animals, such as the reduction of indigenous aquatic animals caused by predation or competition, while p26 focuses on evaluating the potential impact of the introduction of non-native species on the organisms in the waters, such as the large amount of aquatic plant or algae ingested by non-native species, resulting in changes in the water environment, thereby affecting biodiversity, or the introduction of more powerful pathogens, resulting in the reduction of biodiversity. To differentiate, we have made the following modifications in Table 1 and Table 2:
Table 1. The indices of the risk assessment system for the introduction of aquatic animals.
|
Primary Index |
Secondary Index |
Tertiary Index |
|
Hazard assessment of introduced species (R1) |
Self-hazard of introduced species (P2) |
Suitability of environmental factors (including temperature, dissolved oxygen, salinity, pH, etc.) (p21) |
|
Natural enemies of introduced species (p22) |
||
|
Feeding habits of introduced species (p23) |
||
|
Impact of introduced species on indigenous aquatic animals in receiving waters (p24) |
||
|
Impact of introduced species on the abiotic environment of the receiving waters (p25) |
||
|
Impact of introduced species on biodiversity (mainly referring to the impact on algae, aquatic plants, microorganisms, etc.) (p26) |
Table 2. Assessment criteria for the hazard assessment of introduced species
|
Primary Index |
Secondary Index |
Tertiary Index |
Assessment Criteria and Evaluation Bases |
|
0 (Negligible Risk), 1 (Low Risk), 2 (Slight Risk), 3 (Medium Risk), 4 (High Risk), 5 (Extremely High Risk) |
|||
|
R1 |
P2 |
p24 |
(0) The introduced species have little impact on the indigenous aquatic animals in the receiving waters. (1) The introduced species may have a certain negative impact on the indigenous aquatic animals in the receiving waters. (2) The introduced species will have a certain negative impact on the indigenous aquatic animals in the receiving waters, but it can be completely controlled. (3) The introduced species will have a greater negative impact on the indigenous aquatic animals in the receiving waters, but it is still within the controllable range. (4) The introduced species will have a huge negative impact on the indigenous aquatic animals in the receiving waters. (5) The introduced species will have a very serious and irreversible negative impact on the indigenous aquatic animals in the receiving waters. |
|
p26 |
(0) The introduced species hardly have a negative impact on the biodiversity of the region. (1) The introduced species have little negative impact on the biodiversity in the region. (2) The presence of introduced species would pose a potential threat to biodiversity in the region. (3) The introduced species would harm the biodiversity in the region by feeding on algae or aquatic plant in large quantities, carrying harmful organisms, etc., but it is still within the controllable range. (4) The destruction of biodiversity in the region by introducing species has exceeded the self-regulation range of the ecosystem and human control range. (5) Multiple factors would have led to a sharp decrease in biodiversity in the region. |
16. Table 1, p37. Unclear what “all levels” are.
Response: Thank you for your suggestion. “All levels” represent “district level, city level, and provincial level”, and we added the supplementary description after it.
Table 1. The indices of the risk assessment system for the introduction of aquatic animals.
|
Primary Index |
Secondary Index |
Tertiary Index |
|
Entry assessment (R2) |
Official fishery and medical management systems of both countries (P3) |
Status of nationally recognized aquatic animal epidemic laboratories at all levels (district level, city level and provincial level) (p37) |
17. Page 5, section 2.2. Explain what were the experts in your case. How many, how were they selected, and how can this selection bias the results.
Response: Thank you. In this section, we quoted these sentences from “Analytical hierarchy process: revolution and evolution”: One of the most critical features of AHP is the need for accuracy in forming the pairwise comparison matrices defined by the experts. Saaty (1980) described the consistency ratio criterion as an upper limit for each matrix and the hierarchical analytical process to test accuracy. The pairwise comparison matrices must be revised if they display a consistency value higher than a predetermined level. The complexity of the subsequent evaluation process may decrease participants’ motivation and even reduce the accuracy of the results obtained. The original text refers to the person who establishes accurate pairwise comparison matrices as an expert. In this section, it will be better to change it to “users” for better understanding.
Line 167 to 168:
One of the core steps of AHP is to form accurate pairwise comparison matrices defined by the users.
18. Page 14. What field observations? Explain which field observations were performed in your case.
Response: Thank you. We feel sorry that the expression here is not very clear. Our work includes participating in the work of relevant units such as the Ministry of Ecology and Environment, the Ministry of Natural Resources, and the China customs, observation of farms and aquariums with target species (morphological characteristics, lifestyle habits, etc.), and communicate with aquaculture related personnel. In order to help readers to have a better understanding, we have made the following modifications:
Line 254 to 260:
In this assessment, we used the following methods to score various indices: literature reviews, consultation with experts, participating in the work of relevant units such as the Ministry of Ecology and Environment, the Ministry of Natural Resources, and the China customs, observation of farms and aquariums with target species and communicate with aquaculture related personnel. The following table shows the scoring results of each tertiary index of risk assessment. (Among these tertiary indices, some indices require discussion before scoring. See Appendix A for details of these indices.)
19. Page 16. “Pterygoplichthys pardalis has now established large natural populations”. Natural populations? Do you mean wild populations?
Response: Thank you for your suggestion. We agree with you that “Natural populations” here should be “wild populations”.
Line 285 to 288:
However, because of its wide-ranging tolerance, strong reproductive capacity, omnivorous feeding habits, and other characteristics, Pterygoplichthys pardalis has now established large wild populations in most rivers around the world, seriously affecting agricultural production and ecological environments of introduction areas.
20. Page 19. What are “their bodies or reproductive locales”?
Response: Thank you. Sorry about that the expression here was inappropriate. We have made the following modifications:
Line 410 to 413:
The main reason for the migration or invasion of non-native species is that intentional or unintentional actions of humans cause their own individuals or reproductive bodies to spread beyond the limits of the normal geographic regions to which they originally belonged.